# Mutual Jellification of Two Bactericidal Cationic Polymers: Synthesis and Physicochemical Characterization of a New Two-Component Hydrogel

**DOI:** 10.3390/pharmaceutics14112444

**Published:** 2022-11-11

**Authors:** Silvana Alfei, Alessia Zorzoli, Danilo Marimpietri, Anna Maria Schito, Eleonora Russo

**Affiliations:** 1Department of Pharmacy, University of Genoa, Viale Cembrano, 16148 Genoa, Italy; 2Cell Factory, IRCCS Istituto Giannina Gaslini, Via Gerolamo Gaslini 5, 16147 Genoa, Italy; 3Department of Surgical Sciences and Integrated Diagnostics (DISC), University of Genoa, Viale Benedetto XV, 6, 16132 Genova, Italy

**Keywords:** styrene-based bactericidal copolymer (CP1), styrene-based homopolymer (OP2), dose-dependent cytotoxicity studies, human fibroblast, gelling agent, spectroscopic characterization, rheological experiments, swelling and porosity characteristics, kinetic mathematical models

## Abstract

Here, a new two-component hydrogel (CP1OP2-Hgel) was developed, simply by dispersing in water two cationic bactericidal polymers (CP1 and OP2) effective against several multidrug-resistant (MDR) clinical isolates of the most relevant Gram-positive and Gram-negative species. Interestingly, while OP2 acts only as an antibacterial ingredient when in gel, CP1 works as both an antibacterial and a gelling agent. To verify whether it would be worthwhile to use CP1 and OP2 as bioactive ingredients of a new hydrogel supposed for a future treatment of skin infections, dose-dependent cytotoxicity studies with CP1 and OP2 were performed on human fibroblasts for 24 h, before preparing the formulation. Although a significant cytotoxicity at concentrations > 2 µM was evidenced for both polymers, selectivity indices (SIs) over 12 (CP1) and up to six (OP2) were determined, due to the powerful antibacterial properties of the two polymers, thus supporting the rationale for their formulation as a hydrogel. The chemical structure and morphology of CP1OP2-Hgel were investigated by PCA-assisted attenuated total reflectance (ATR) Fourier-transform infrared (FTIR) analysis and scanning electron microscopy (SEM), while its rheological properties were assessed by determining its dynamic viscosity. The cumulative weight loss and swelling percentage curves, the porosity, and the maximum swelling capability of CP1OP2-Hgel were also determined and reported. Overall, due to the potent bactericidal effects of CP1 and OP2 and their favorable selectivity indices against several MDR pathogens, good rheological properties, high porosity, and strong swelling capability, CP1OP2-Hgel may, in the future, become a new weapon for treating severe nosocomial skin infections or infected chronic wounds. Further investigations in this sense are currently being carried out.

## 1. Introduction

Frequently touched objects in the hospital environment can be a dangerous source of bacterial infections, mainly of the skin, to healthcare professionals, patients, and visitors [1].

Skin and soft tissue infections (SSTIs) [2] or acute bacterial skin and skin structure infections (ABSSSIs) [3] are diseases of the skin and associated soft tissues (such as loose connective tissue and mucous membranes) caused by bacterial species, which require treatment with antibiotics.

Until 2008, two categories of skin infections were recognized consisting of complicated SSSIs (cSSSIs) and uncomplicated SSSIs (uSSSIs) [4], which had different regulatory approval requirements [3]. In particular, uSSSIs included simple abscesses, impetiginous lesions, furuncles, and cellulitis [3], while cSSSIs included infections involving deeper soft tissue or requiring significant surgery, such as infected ulcers, burns, and major abscesses or a significant underlying disease state that complicates the response to treatment [3]. However, when localized to particular anatomical sites, such as the rectal area, where the risk of involvement of anaerobic or Gram-negative pathogens is greater, superficial infections or abscesses have also been considered complicated infections [3]. uSSSIs are most often caused by *Staphylococcus aureus* and *Streptococcus pyogenes*, while cSSSIs can also be caused by a number of other pathogens [3]. Concerning the diagnosis of SSSIs it has been observed that physicians are generally not used to culture methods for identifying the pathogen [5]. Moreover, common treatments are often empirical, consisting in the choice of antibiotics based on the symptoms reported by patients and the location of the infection. Furthermore, treatment strategies are often designed based on trial and previous errors [5]. Unfortunately, to treat SSSIs, physicians often prescribe broad-spectrum antibiotics, a questionable practice that contributes to the increasing emergence of antibiotic resistance, a trend linked to the widespread and unjustified use of these drugs. The increased prevalence of antibiotic resistance is evident for methicillin-resistant *Staphylococcus aureus* (MRSA), commonly involved in SSSIs, whose prognosis is worsened, and treatment options limited, by its presence. For less severe infections, microbiological assessment of tissue culture has been shown to be of great use in guiding management decisions [5]. There is no evidence to support or disagree with the use of Chinese herbal medicines in the treatment of SSTIs [6]. Concerning the armamentarium currently available for the treatment of skin infections, linezolid is an antibiotic used in the hospital setting for infections caused by multidrug-resistant (MDR) Gram-positive bacteria [7,8], including streptococci, vancomycin-resistant enterococci (VRE), and MRSA [7,9]. Although it can be used for a variety of other infections, including drug-resistant tuberculosis, linezolid is primarily used to treat skin infections and pneumonia [8,10]. It can be administered both intravenously and orally [8]. When given for short periods, linezolid is a relatively safe antibiotic [11], but used for longer periods it may cause nerve damage, including optic nerve damage, which may be irreversible [12]. Unfortunately, although bacterial resistance to linezolid remained low until 2014 [13], many clinical isolates have been found resistant to linezolid in the last decade. Furthermore, linezolid has been shown to be clinically ineffective against most Gram-negative bacteria, including those of the genus *Pseudomonas* and *Enterobacteriaceae* [14].

Tedizolid (formerly torezolid, trade name Sivextro) [15] is an antibiotic of the oxazolidinone class, usually administered as a prodrug of phosphate esters, 4 to 16 times more potent than linezolid against staphylococci and enterococci [16]. It is clinically approved for the treatment of cSSSIs [17] caused by MRSA and methicillin-sensitive strains (MSSAs), *S. pyogenes, S. agalactiae*, *S. anginosus*, *S. intermedius*, *S. constellatus*, and *Enterococcus faecalis* [18,19,20,21]. Although the most common side effects are generally of mild or moderate severity, including nausea (feeling sick), headache, diarrhea, and vomiting [22], the use of this drug to treat ABSSSIs is recommended only in adults [22].

Due to this scenario that shows limited resources to treat severe skin infections, and especially those sustained by Gram-negative MDR pathogens, new effective antibacterial agents are urgently needed. Furthermore, concerning the possible formulation of new bactericidal/antibacterial compounds in potentially clinically applicable dosage forms, their conversion into hydrogels could effectively satisfy the urgent demand for new and effective antibacterial formulations suitable for topical administration [23]. Hydrogels are 3D polymer networks generally crosslinked by either physical interactions or covalent bonds [24]. Currently, hydrogels with an antibacterial function are of great interest in biomedical research. Many advanced antibacterial hydrogels have been and are being developed, each one with unique qualities, such as a high-water swelling ability, high oxygen permeability, improved biocompatibility, ease of drug loading and release, and structural diversity [25]. In this regard, Andrèn et al. recently developed a non-toxic and hydrolytically rapidly degradable antibacterial hydrogel for the preventive treatment of surgical site infections during the first crucial 24 h period without relying on conventional antibiotics [26]. Moreover, hydrogels consisting of cationic polymer chains obtainable either by physical or chemical crosslinking have been extensively studied as an alternative material for antibacterial applications [25]. In this context, aiming to develop new bactericidal formulations for topical use, we recently developed a Bingham pseudoplastic hydrogel (CP_1.1-Hgel). Interestingly, it was achieved simply by dispersing in water a non-crosslinked cationic styrene-based copolymer (CP1) endowed with potent bactericidal effects and unexpectedly capable of self-forming gels, without the use of gelling agents or other additives. Although more in-depth investigations are currently underway to evaluate its effective use for the treatment of skin infections, the first studies carried out on CP_1.1-Hgel and the absence of additives (potentially incompatible with the skin or capable of interfering with the bioactivity of CP1) supported its further development. Here, to expand our research in the field, we have reported the development of a new two-component hydrogel, which, due to its physicochemical characteristics, and the biological profiles of its ingredients, could hold promise for the treatment of bacterial-borne skin infections sustained by bacteria with a difficult resistance profile, or for improving infected chronic wounds.

In particular, to prepare the new gel, we used two recently reported styrene-based polymers (namely CP1 and OP2, Figure 1) endowed with rapid bactericidal effects and very low minimum inhibitory concentrations (MICs) (0.1–0.8 µM for CP1 and 0.35–2.9 µM for OP2) [27] against several MDR clinical isolates, including those mostly found in hospital settings and that are responsible for severe skin infections. The new two-component hydrogel based on both CP1 and OP2, different to our first experience with CP1, proved to have different rheological properties, showing a dilatant shear thickening behavior. Moreover, it was characterized by a very high porosity and swelling ability. Interestingly, even if not cross-linked, copolymer CP1 (Figure 1a) takes part in the gel formulation both as a bioactive ingredient like homopolymer OP2 (Figure 1b), and, due to its capability of self-forming gels, also as a thickening agent for gelling OP2.

Practically, the CP1OP2-Hgel developed here was obtained upon simple dispersion of the two polymers in water, without the need of gelling agents or other additives, thus limiting possible skin incompatibilities in future topical administrations or undesired interactions between components, as well as an unexpected reduction in the antibacterial effects of CP1 and/or OP2. Therefore, CP1OP2-Hgel may possibly merge the antibacterial characteristics of the two ingredients, thus enlarging the spectrum of action of single components. Biological experiments to evaluate possible enhancements or reductions in the antibacterial effects of CP1 and OP2 by their formulation in gel, as well as to assess the antibiofilm efficacy of CP1OP2-Hgel, are currently being undertaken. However, before preparing the formulation, we checked whether it was worth using CP1 and OP2 as bioactive ingredients of a new hydrogel hypothesized for a future skin use, by carrying out dose-dependent cytotoxicity studies with CP1 and OP2 on human fibroblasts at 24 h of exposure. Following results that confirmed the rationale for their formulation as a gel, CP1OP2-Hgel was prepared and characterized by ATR-FTIR spectroscopy and scanning electron microscopy (SEM) to investigate its chemical structure and morphology. Its rheological properties were studied by determining its dynamic viscosity and processing data by proper mathematical models; the water loss profile over time, the equilibrium swelling ratio percentage, and the maximum swelling capacity and porosity were obtained by carrying out the appropriate experimental tests, as described in the work. Finally, although the NH_2_ content in CP1 and OP2 as isolated polymers had already been evaluated [27], CP1OP2-Hgel was titrated here potentiometrically to determine the NH_2_ equivalents per gram of gel.

## 2. Materials and Methods

### 2.1. Chemicals and Instruments

The styrene-based cationic monomers M1 (4-vinyl-benzyl-ammonium chloride) and M2 (2-(4-vinyl-phenyl)-ethyl-ammonium chloride), which are necessary to prepare copolymer CP1 and homopolymer OP2 used in this work to form CP1OP2-Hgel and in turn CP1 and OP2 (Figure 1), were prepared according to procedures recently described [27]. In particular, CP1 was prepared by the radical solution copolymerization of M1 with dimethylacrylamide (DMAA), while OP2 was prepared by the radical solution homopolymerization of M2 [27]. All reagents and solvents were from Merck (formerly Sigma-Aldrich, Darmstadt, Germany) and were purified by standard procedures. Azo-*bis*-isobutyronitrile (AIBN) used to reproduce the polymerization reactions and prepare CP1 and OP2 was crystallized from methanol (MeOH). Organic solutions were dried over anhydrous magnesium sulfate and were evaporated using a rotatory evaporator operating at a reduced pressure of about 10–20 mmHg. The melting ranges of the solid compounds in this study were determined on a 360 D melting point device, resolution 0.1° C (MICROTECH S.R.L., Pozzuoli, Naples, Italy). Melting points and boiling points are uncorrected. The ATR-FTIR analyses were recorded on a Spectrum Two FT-IR Spectrometer (PerkinElmer, Inc., Waltham, MA, USA). Scanning electron microscopy (SEM) images were obtained with a Leo Stereoscan 440 instrument (LEO Electron Microscopy Inc., Thornwood, New York, NY, USA). Potentiometric titrations were carried out using a Hanna Micro-processor Bench pH Meter (Hanna Instruments Italia srl, Ronchi di Villafranca Padovana, Padova, Italy), which was calibrated using standard solutions at pH = 4, 7, and 10, before titrations. Column chromatography carried out to purify the intermediates during the synthesis of M1 and M2 were performed using appropriate glass columns and Merck (Darmstadt, Germany) silica gel (70–230 mesh). Similarly, thin layer chromatography (TLC) was carried out using aluminum-backed silica gel plates (Merck DC-Alufolien Kieselgel 60 F254, Merck, Washington, DC, USA), and detection of spots was made by UV light (254 nm), using a Handheld UV Lamp, LW/SW, 6W, UVGL-58 (Science Company^®^, Lakewood, CO, USA). For the rheological characterization of the gel, a Brookfield viscometer (Viscostar-R, Fungilab S.A., Torino, Italy) was used.

### 2.2. Dose-Dependent Cytotoxicity Experiments with CP1 and OP2

#### 2.2.1. Human Fibroblast Isolation and Culture

Human primary fibroblasts were generated from the skin flap of a previous cesarean scar in women undergoing cesarean delivery at the Obstetrics and Gynecology Unit of the G. Gaslini Institute, Genova, Italy. Written informed consent was previously obtained. The skin sample was processed under a sterile laminar flow cabinet 4 h after collection. Briefly, 5–10 cm of skin was washed twice with PBS (Dulbecco’s Phosphate-Buffered Saline *w/o* Calcium and Magnesium, Euroclone S.p.A., Milan, Italy) to remove contaminant red blood cells, transferred in a 90–100 mm Petri dish (Corning) and cut into ~5 mm segments using sterile scissors. Fat tissue was removed. About 3–4 Petri containing 10 pieces each were plated and left to dry for 10 min to allow fragment adhesion to the plastic surface. Then, 10 mL of DMEM High Glucose (Dulbecco’s modified Eagle medium) supplemented with 10% fetal bovine serum (*v*/*v*) (FBS), 1% penicillin–streptomycin, and 1% glutamine (Euroclone S.p.A., Milan, Italy) was added to each Petri and incubated at 37° in a humidified atmosphere with 5% CO_2_. At day 9, skin fragments were removed, and fibroblasts were allowed to expand. Fresh culture medium was added every 4 days. At day 20, fibroblasts were detached using TrypLE Cell Therapy System CTS (Gibco, Thermo Fisher Scientific, Life Technologies, Milano Brianza, Italy) and collected for count and vitality assay by Trypan Blue (Sigma Aldrich, Milan, Italy). An additional expansion passage was performed in T-75 cm^2^ plastic flasks (Falcon, Corning, Glendale, AZ, USA) and, after 7 days, cells were finally collected, counted, and frozen. Phenotypic characterization of fibroblast was performed using anti-human Cadherin11 APC (BioLegend, San Diego, CA, USA) using Gallios^®^ flow-cytometer (Beckman Coulter, Rome, Italy); data not shown.

#### 2.2.2. Viability Assay

Fibroblasts were thawed, expanded, and seeded in 96-well plates (at 4 × 10^3^ cells/well) in complete medium for 24 h. The seeding medium was removed and replaced with fresh complete medium supplemented with increasing concentrations of CP1 (0.01–20 μM) and OP2 (0.1–40 μM). Cells (quadruplicate samples for each condition) were then incubated for an additional 4 h and, subsequently, for 24 h. The effect on cell growth was evaluated by a fluorescence-based proliferation and cytotoxicity assay (CyQUANT^®^ Direct Cell Proliferation Assay, Thermo Fisher Scientific, Life Technologies, Milano Brianza, Italy) according to the manufacturer’s instructions. Briefly, at each time point, an equal volume of detection reagent was added to each well and incubated for 60 min at 37 °C. The fluorescence signal was measured using the monochromator-based M200 plate reader (Tecan, Männedorf, Switzerland) set at 480/535 nm.

### 2.3. Determination of Selectivity Indices (SIs) for CP1 and OP2

From the experiments described in Section 2.2.2, we determined the *LD_50_*.values at 24 h of exposure of MAIL-2 cells to CP1 and OP2. The values of *LD_50_* obtained and the MIC values (also determined at 24 h of exposure) previously reported for CP1 and OP2 [27] were used to calculate their *SI* values (*SIs*) with respect to each isolate on which the polymers were tested, according to Equation (1).
*SI**=**LD_50_**/**MIC*(1)
where *LD_50_* is the dose of CP1 or OP2 that was able to halve the percentage of cell viability at 4 h of exposure, while MIC is the minimum dose of CP1 and OP2 that was able to inhibit the growth of bacteria.

### 2.4. CP1OP2-Hgel Preparation

OP2 (0.25 mL, 80.5 mg; 1.8 µmol) was weighted in a graduated centrifuge tube (Ø_est_ = 14 mm, V = 10 mL) and added with equimolar CP1 (0.35 mL, 284.5 mg, 1.8 µmol), obtaining a total volume of 0.6 mL (Vi) and a total weight of 365.0 mg (Wi). The average molecular weights used to calculate the µmol of CP1 and OP2 corresponding to their weights were viscosity average MMs (Mv) determined as reported in our previous study [27]. Then, at room temperature and under magnetic stirring, deionized water was added (9.4 mL) up to a total volume of 10 mL. When an homogeneous mixture was observed, the magnetic stirrer was removed and the dispersion was sonicated at 37 °C for 15 + 15 min, and then degassed for 9 min, using an Ultrasonic Cleaner 220 V, working at a frequency of 35 kHz, timer range 1–99 min, temperature range 20 to 69 °C (68 to 156 °F) (VWR, Milan, Italy). The dispersion was then centrifugated at 4000 rpm for 15 min to separate the gel at its maximum content of water and maximum swelling capacity from water in excess, which was removed. The tube was then turned upside down on filter paper to remove residual water and left for 10 min. The weight of the obtained CP1OP2-Hgel was 4.1362 g (Wf), while its volume was 4.10 mL (Vf), corresponding to the weight and the volume of gel at its maximum swelling capability. Accordingly, the volume of water in which CP1 and OP2 were finally dispersed, once the fraction of water not absorbed was removed, was 3.5 mL. The concentration of the two polymers was 23.0 mg/mL (516.6 µM) for OP2 and 81.3 mg/mL (516.8 µM) for CP1 (8.8% CP1+OP2 wt/wt). The initial volume (Vi) and the final volume (Vf), as well as the initial weight (Wi) and the final weight (Wf), were used to determinate the maximum swelling capability percentages (S%) and porosity percentages (P%) by volumes and weights, respectively, according to Equations (2a), (2b), (3a) and (3b).
P (%) = 100 × (Vf − Vi)/Vf(2a)
P (%) = 100 × (Wf − Wi)/Wf(2b)
S (%) = 100 × (Vf − Vi)/Vi(3a)
S (%) = 100 × (Wf − Wi)/Wi (3b)

CP1OP2-Hgel was then left in the carefully sealed tube to prevent water evaporation and stored in the fridge for subsequent characterization experiments, including ATR-FTIR, water loss, potentiometric titrations, and rhelogical studies. A fraction of the prepared CP1OP2-Hgel was lyophilized to complete its characterization by SEM analyses.

### 2.5. ATR-FTIR Spectra

ATR-FTIR analyses were carried out both on the soaked and on the dried gel directly on the samples. The spectra were acquired from 4000 to 600 cm^−1^, with 1 cm^−1^ spectral resolution, co-adding 32 interferograms, with a measurement accuracy in the frequency data at each measured point of 0.01 cm^−1^, due to the internal laser reference of the instrument. Acquisitions were made in triplicate, and the spectra shown in Section 3.3 are the most representative images. The spectral data obtained were then included with those of CP1, OP2 [27], and those of the soaked and dry CP1-based gel, whose ATR-FTIR spectra are available in Appendix A, in a data set matrix, and were processed using the principal component analysis (PCA) by means of PAST statistical software (paleontological statistics software package for education and data analysis, freely downloadable online at: https://past.en.lo4d.com/windows; accessed on 29 October 2022). In particular, we arranged the FTIR data of the six spectra in a matrix 3401 × 6 (*n* = 20,406) of measurable variables. For each sample, the variables consisted of the values of transmittance (%) associated with the wavenumbers (3401) in the range 4000–600 cm^−1^. The spectral data in the matrix were pretreated by autoscaling.

### 2.6. Scanning Electron Microscopy (SEM)

The microstructure of the lyophilized hydrogel was investigated by SEM analysis. In the performed experiments, the hydrogel was first frozen by fast immersion into liquid nitrogen. Later, the sample was lyophilized at −55 °C for 24 h, using a freeze drying system (Labconco, Kansas City, MI, USA). Then, it was broken in liquid nitrogen, fixed on aluminum pin stubs, and sputter-coated with a gold layer of 30 mA for 1 min, to improve the conductivity, and an accelerating voltage of 20 kV was used for the sample’s examination. The micrographs were recorded digitally using a DISS 5 digital image acquisition system (Point Electronic GmbH, Halle, Germany).

### 2.7. Weight Loss (Water Loss) Determination

A sample of CP1OP2-Hgel was deposited with a spatula into a glass Petri capsule. The deposited gel was weighed (198.4 mg). The glass capsule was then placed in a temperature-controlled oven (37 °C), and water loss was monitored as a function of time until a constant weight was reached (24 h). The cumulative water loss percentage was determined using Equation (4):(4)$$Weight\;loss\;\left(\%\right)=\left(MQ-Mt\right)/MQ\times 100$$
where *MQ* and *Mt* are the initial hydrogel mass and hydrogel mass after a time *t*, respectively.

### 2.8. Swelling Rate

The swelling measurements were carried out in a centrifuge tube, where 9.5 mg of the dried gel obtained in the previous experiment was inserted and added with an excess of water. At fixed intervals of time (15, 30, 45, and 60 min), the swollen gel was precipitated by centrifugation at 4000 rpm for 15 min, and the water not absorbed by the sample was removed by inverting the tube on filter paper and letting it rest for 10 min. After removing the residual water on the tube walls with a cotton swab, we proceeded to weight the swollen gel. The cumulative swelling ratio percentage (*Q*%) as a function of time was calculated from Equation (5):(5)$$Q\;\left(\%\right)=\left(Ws-Wd\right)/Wd\times 100$$
where *Wd* and *Ws* are the weights of the lyophilized and swollen gel, respectively. The equilibrium swelling ratio (*Q*_equil_) was determined at the point the hydrated gels reached a constant weight.

### 2.9. Potentiometric Titration of CP1OP2-Hgel

Potentiometric titrations were performed at room temperature and the titration curves of CP1OP2-Hgel were obtained. In a representative experiment, an exact amount of gel (284.1 mg, 25.0 mg CP1 + OP2, 19.5 mg CP1, and 5.5 mg OP2) was diluted with 30 mL of Milli-Q water (m-Q). Then, it was added with a standard 0.1 N NaOH aqueous solution (3.0 mL), under magnetic stirring, to reach a strongly basic pH value (pH = 10.42). The solution was potentiometrically titrated under stirring by adding 0.2 mL aliquots of a standard 0.1 N HCl aqueous solution up to 3.0 mL, 0.5 mL, 6.0 mL, and, finally, 1.0 mL, up to a total of 10.0 mL, and measuring the pH values of the resulting solutions [27,28]. Titrations were performed in triplicate and measurements were reported as mean ± SD. The titration curve shown in the Section 3 is that obtained by plotting the data obtained by carrying out the representative experiment described here.

### 2.10. Rheological Studies

The rheological properties of CP1OP2-Hgel were assayed by a continuous shear method using a Brookfield viscometer (Viscostar-R, Fungilab S.A., Torino, Italy). In particular, samples of about 2 g were subjected to shear rates ranging from 1 to 100 s^−1^. All measurements were carried out at room temperature and were expressed as the mean values of five independent determinations. The image reported in the Section 3 is representative of one determination.

### 2.11. Statistical Analyses

The statistical significance of the differences between the experimental and control groups in cytotoxicity studies was determined by a two-way analysis of variance (ANOVA) with Bonferroni correction. The analyses were performed with Prism 5 software (GraphPad, La Jolla, CA, USA). Asterisks indicate the following *p*-value ranges: * = *p* < 0.05, ** = *p* < 0.01, *** = *p* < 0.001.

## 3. Results and Discussion

As reported in our previous work, both CP1 and OP2 demonstrated strong bactericidal properties against several MDR clinical isolates of different species of Gram-positive and Gram-negative pathogens, regardless of their complex profiles of resistance (MICs = 0.1–0.8 µM (CP1) and 0.34–2.8 µM (OP2)) [27]. Importantly, when tested with time–kill experiments on different clinically relevant isolates of the Gram-positive and Gram-negative species, such as MRSA, *E. coli*, and *P. aeruginosa* strains, both cationic polymers, and especially OP2, demonstrated powerful and rapid bactericidal effects at 4 x MICs, regardless of the resistance profiles of the bacteria tested [27]. We also observed that, in some cases, their antibacterial effects were complementary. For example, while OP2 was remarkably active on the three most relevant species of non-fermenting Gram-negative bacteria, such as *A. baumannii*, *S. malthophilia*, and *P. aeruginosa*, CP1, although with lower MICs, was also active on *P. aeruginosa*, while it was ineffective on *A. baumannii* and *S. malthophilia.* Interestingly, we observed that CP1 self-formed gels by its simple dispersion in water without the use of gelling agents or other additives. On these findings, we have recently prepared and completely characterized a CP1-based self-formed hydrogel (CP1_1.1-Hgel), obtained by the simple dispersion in water of CP1 at 1.1% wt/wt. Following the characterization results and the biological behavior of its unique ingredient (CP1) on bacteria [27] and on human fibroblasts at 4 h of exposure (Appendix A), this first gel developed by us appears interesting for the development of a new antibacterial gel formulation topically applicable to treat skin infections. In this study, it was speculated that CP1 could be used as a gelling agent to formulate OP2 in a new two-component hydrogel that, thanks to the bactericidal properties of the two ingredients, could have a broader spectrum of action. However, before going ahead with the formulation, we assessed first the cytotoxic behavior of OP2 on human fibroblasts, as previously done for CP1, exposing cells to the polymer for 4 h. Then, to obtain data more compatible with MICs, which were determined at 24 h of exposure, as recommended in microbiology, we evaluated the cytotoxic behavior of both CP1 and OP2 on fibroblasts after 24 h of exposure.

### 3.1. Dose-Dependent Cytotoxicity Experiments

Assuming a possible cutaneous use of the CP1OP2-based gel as an antibacterial agent, due to the potent bactericidal effects of its ingredients, dose-dependent cytotoxicity studies were first carried out with OP2 on human fibroblasts (MAIL-2) at 4 h of exposure, as already done for CP1 (Appendix A). Subsequently, we performed the same experiments exposing the MAIL-2 cells to both CP1 and OP2 for 24 h. Appendix A shows the bar graph of the cells’ viability percentage vs. OP2 concentration after 4 h of exposure. Here, Figure 2 and Figure 3 show the same bar graphs obtained by the dose-dependent cytotoxicity experiments carried out for 24 h of exposure of MAIL-2 cells to CP1 and OP2, respectively.

Human fibroblasts were selected for cytotoxicity experiments because they are widely used for the evaluation of the cytotoxicity of compounds intended for topical administration as cosmetic formulations [29,30]. The cytotoxicity of OP2 and CP1 was first evaluated at four hours of exposition because CP1 and OP2 displayed bactericidal activity at 4 and 2 h of exposure, respectively [27]. The cytotoxicity of both CP1 and OP2 was also assessed up to 24 h, to obtain their LD_50_ values at 24 h, which were necessary to calculate the values of the selectivity indices according to Equation (1), using MICs, which were determined at 24 h of exposure as well.

Considering the cytotoxicity of both CP1 and OP2 after 24 h of exposure, CP1 was not cytotoxic for concentrations < 1 μM; cell viability significantly higher than 50% was observed for 1μM CP1 concentrations, whereas CP1 was highly cytotoxic at concentrations ≥ 2 μM (Figure 2). OP2 was less cytotoxic than CP1, since cell viability significantly higher than 50% was observed for OP2 concentrations < 2 μM. Cells viability was of 50% for 2 μM OP2 concentrations, while it was highly cytotoxic at concentration over 2 μM. By a comparison between the cytotoxic profiles of CP1 and OP2 obtained at 4 h and those obtained at 24 h, it can be asserted that no significant difference in the cell viability was observed, thus establishing that the cytotoxicity of both polymers is only dose-dependent, and it is not dependent on time of exposure. To obtain more representative information on the feasibility of using CP1 and OP2 in a gel formulation that is possibly applicable to the skin, we determined their SIs against several clinically relevant MDR pathogens, including strains responsible for severe skin infections. Note that SI values are important in deciding the therapeutic applicability of a new antibacterial agent because they measure its capability to selectively inhibit the bacterial cell without damaging the eukaryotic one. To calculate the desired SI values, we first reported the data of cell viability % vs. the concentrations of CP1 and OP2, obtaining the curves shown in Figure 4a and Figure 5a, respectively.

The cell viability (%) reported in the graphs for each concentration point was the average of the results obtained from four independent determinations. Using the points of the curve in Figure 4a up to a concentration of 5 μM and of those of the curve in Figure 5a up to a concentration of 20 μM, the correspondent dispersion graphs were obtained, and the best fitting tendency lines were reached using the tool provided by Microsoft Excel software. We used only the described concentrations, because over such concentrations, cell viability remained unchanged, and viable cells were insignificant. The tendency line was polynomial for CP1 and logarithmic for OP2. Their good fitting with the dispersion graphs was assured by the very high values of the coefficients of correlation (R^2^). The equations of the tendency lines were used to determine the desired LD_50_. Figure 4b and Figure 5b show the used dispersion graphs, the best fitting tendency lines, and the related equations used to compute the LD_50_ of CP1 and OP2. The obtained equations, the R^2^ values, the computed values of LD_50_, and the range of SIs obtained for CP1 and OP2 calculated according to Equation (1) are reported in Table 1.

The SI values of CP1 and OP2 calculated for each isolate used in the previous study to determine the MICs of CP1 and OP2 [27] are instead reported in Table 2.

According to Table 1, the LD_50_ of CP1 was slightly >1 µM, while that of OP2 was >2 µM, but being the concentration of CP1 necessary to inhibit all Gram-positive and Gram-negative MDR clinical isolates < 1 µM (Table 2), the relative SI values were all >1 up to values of 12. Similarly, with the exception of *K. pneumoniae* 509, for which the OP2 concentration required to inhibit all Gram-positive and Gram-negative MDR clinical isolates was <2 µM (Table 2), their SI values were all >1, as for CP1, and up to values of 6. In particular, the SI range for CP1 and OP2 was 1.5–12 and 1.5–6, respectively, against isolates of Gram-positive species and 1.5–3.0 and 0.75–3.0, respectively, against Gram-negative strains. Since these values are considered acceptable for a hypothetical, clinically applicable, new antibacterial agent [31,32,33,34], we continued to use CP1 and OP2 as ingredients to develop a new bactericidal gel with potential for topical administration.

### 3.2. CP1OP2-HGel Preparation

The potent antibacterial effects observed for both CP1 and OP2 against several alarming MDR clinical isolates responsible for severe nosocomial infections that are nearly untreatable inspired us to formulate them in a new antibacterial dosage form possibly suitable for cutaneous use. Since CP1 was shown to self-form hydrogels [27] and was successful in providing a single-component hydrogel with excellent physicochemical characteristics, which will soon be published, we thought to formulate them in a new bicomponent hydrogel by exploiting CP1 both as a bactericidal ingredient and as a gelling agent. This strategy would have allowed us to avoid the use of additional additives, which could cause unwanted side reactions both in the preparation phase and in future, when applied to skin. Hydrogels consist of three-dimensional network structures obtained from synthetic and/or natural polymers capable of absorbing and retaining significant amounts of water [35]. Hydrogels obtained from several polymers have been produced and employed in tissue engineering, pharmaceutical, and biomedical fields [36]. Therefore, also supported by the favorable selectivity index values determined for CP1 and OP2, according to the procedure described in Section 2.4, we prepared a CP1OP-Hgel containing the maximum amount of water which the equimolar combination of CP1 and OP2 was capable of absorbing. Table 3 collects the experimental data of CP1OP2-Hgel preparation, as well as details about CP1 and OP2 concentrations in the prepared hydrogel.

An image is available in the Appendix A that shows the very viscous hydrogel obtained, whose volume and weight corresponded to those of the gel at its maximum swelling capability.

Using Equations (2a), (2b), (3a), and (3b), the maximum swelling capability percentages and porosity percentages by volume and by weight were calculated and are reported in Table 3 (last two columns). The porosity of CP1OP2-Hgel by volume and by weight was similar and very high (85% by volume and 91% by weight), while the maximum swelling capability percentages were 583% and 1033%, respectively. In particular, the porosity was 1.2-fold higher than that of the CP1-based gel, while the maximum swelling capability was 1.9–3.4-fold higher. According to what has been reported, the high porosity and remarkable capacity to absorb water should give CP1OP2-Hgel a high potentiality to function also as a wound-healing hydrogel [37,38]. Obviously, further experiments are necessary to confirm this early hypothesis.

### 3.3. ATR-FTIR Spectra

ATR-FTIR analyses were performed directly on samples of the soaked gel and of the rubbery solid obtained by depositing the gel on a glass slide and heating it to dryness. Spectra were acquired in triplicate and those reported in Figure 6 are representative images.

As expected, although very small bands indicating the presence of the two polymers were detectable at 1509, 1408, 1254, and 1152 cm^−1^ (aromatic C=C and C-N stretching bands), the ATR-FTIR spectrum of the soaked gel (blue line) showed mainly the typical bands of water that are present at about 91.2% wt. with respect to CP1 and OP2, whose combination in the gel represents 8.8% of weight. On the contrary, the spectrum of the dried sample, obtained by heating the soaked gel until a rubbery solid (red line) was obtained, showed mainly the typical bands observable in the spectra of copolymer CP1 and homopolymer OP2 [27], thus establishing that the interactions occurred during the gel formation are reversible, as previously observed for CP1_1.1-Hgel. We assumed an impurity caused the small band at 1739 cm^−1^, probably due to the glass slide used or to residual solvents on the acquisition site, generally indicating the presence of C=O groups of the ester type and not compatible with the structure of ingredients in CP1OP2-Hgel.

#### Principal Component Analysis (PCA) of ATR-FTIR Data

A matrix of over 20,000 variables was constructed collecting the spectral data (wavenumbers) of CP1, OP2, the swollen and dry CP1-based gel, as well as of the swollen and dry CP1OP2-based gel developed here. Then, the resulting dataset was first pretreated by autoscaling and then processed using the PCA, which is an intriguing chemometric tool capable of extracting the essential information from an intricate and enormous set of variables [39,40,41,42,43]. In the present case, the PCA allowed us to visualize the reciprocal positions that the six samples occupied in the scores plots of PC1 vs. PC2 (Figure 7), depending on the presence of similarities or differences in their chemical composition.

As observed, while on PC2 the two types of swollen gels were located at negative scores and close to each other, dried samples and original CP1 and OP2 were positioned at positive values. Both the dried samples (CP1gelD and CP1OP2gelD), as well as CP1 and OP2, were all distant from the swollen gels (CP1gelS and CP1OP2gelS), thus showing the very different composition of the latter with respect to the other four compounds, due to the presence of a high concentration of water (99% in the CP1-based gel and 91% in the CP1OP2-based gel). The minor distance of the CP1OP2gelS from the dried samples confirmed the higher concentration of CP1 and OP2 and the lower content of water with respect to those of CP1gelS. Moreover, the CP1 copolymer and CP1gelD were practically overlapped, thus showing an identical composition and that, upon heating, all water was removed and the original CP1 was reobtained. On the other hand, CP1OP2gelD was located at lower scores and closer to the swollen samples, indicating that, during heating to dryness, a certain percentage of water was maintained. Finally, the position of CP1OP2-based gels closer to CP1 than to OP2 indicated the higher concentration in weight of CP1 with respect to OP2.

### 3.4. Scanning Electron Microscopy (SEM)

The microstructure of the lyophilized hydrogel was investigated by SEM analysis. The appearance of the lyophilized hydrogel is shown in Figure 8a, while a representative SEM micrograph is shown in Figure 8b. SEM images of the lyophilized cationic CP1OP2-Hgel revealed sticky-like particles with spheroidal morphology, high polydispersity, and very different sizes, as expected, since polymers that have particles with very different sizes were used to produce the gel. The most represented size was of about 1 micron, a slightly larger size than that previously determined for CP1 by dynamic light scattering analyses [27], but very small particles with dimensions around 100 nm were observed in the rounded areas of the micrograph.

### 3.5. Weight Loss (Water Loss)

In the Appendix A, Appendix A shows the appearance of CP1OP2-Hgel at time t_0_ of the experiments carried out to monitor water loss over time (Appendix A) and its appearance when fully dried (Appendix A). Appendix A instead collects the data of the experiments, including the weights of the gel at times T_0_-T_10_ and the computed cumulative weight loss ratio percentages. Figure 9a shows the curve obtained when graphing the cumulative weight loss percentages vs. times, while Figure 9b shows the first order kinetic model that, among other mathematical kinetic models, best fit the data of the cumulative weight loss curve.

The water loss from the CP1OP2-Hgel was slightly inferior but much slower than that observed for the gel previously prepared with CP1 only (98.3% at 6 h vs. 94.5% at 24 h). Probably, due to the much higher concentration of the cationic macromolecules in the gel, more hydrogen bonds between the water and heteroatoms present in the copolymer structure were possible, thus causing a higher retention of water. To exactly know the main mechanisms that govern the loss of water from the CP1OP2-Hgel, the data of the curve in Figure 9a were fit with several mathematical kinetic models as reported in other published studies [44,45,46,47], obtaining the related dispersion graphs. The mathematical model which provided the graph, whose linear tendency line (supplied by Microsoft Excel software using the least-squares method) had the highest value of the coefficient of determination (R^2^) was selected as that which best fit the water release data. The mathematical models were tested, and the obtained R^2^ values are reported in Table 4. Accordingly, it was established that the water loss for the CP1OP2-Hgel best fit the first-order kinetic model (Figure 9b), thus establishing a further difference with respect to the CP1-based gel, which fit the Korsmeyer–Peppas kinetic model, as shown in Appendix A.

As reported in the literature, first-order release kinetics state that the change in concentration with respect to the change over time is dependent only on the residual concentration of the compound released. In the present case, the release of water and weight loss over time depended only on the residual concentration of water after the heating periods [46]. The first-order kinetics are described by Equation (6).
(6)$$LogQt=\left(\frac{K}{2.303}\right)*x+LogQo$$
where *Qt* is the amount of water released on time *t*, *Q_o_* is the initial amount (%) of water in the gel, and *K* is the first-order constant. Accordingly, in our case, *K*/2.303 corresponded to the slope of the equation of the linear regression of our first-order mathematical model shown in Figure 9b, while *LogQ_0_* was its intercept. Consequently, the first-order constant was negative and equal to −0.1237 and the original percentage content of water in the weighted CP1OP2-Hgel was 94%.

### 3.6. Cumulative Swelling Ratio Percentages

As described in Section 2.8, swelling measurements were made at times T_0_–T_4_ (0–60 min), until the weight of the swollen gels was approximately constant.

Figure 10 shows the cumulative swelling ratio percentages as a function of time.

The equilibrium swelling ratio (Q_equil_), which was determined at the point where the hydrated gels reached a constant weight, was 2789.3 ± 2.05% (value obtained making the mean of values obtained at times T_2_–T_4_) and was reached immediately after 30 min of hydration. Curiously, the maximum swelling capability (%) of the dried mixture of CP1 and OP2 obtained by heating the prepared gel (24 h, 37 °C) and corresponding to the maximum quantity of water that the dried gel was capable to absorb, was 2.7-fold higher than that absorbed for the original mixture of CP1 and OP2 used to prepare the gel (2789% vs. 1033%).

### 3.7. Potentiometric Titration of CP1OP2-Hgel

Potentiometric titrations of the gel were performed to ensure that the protonation profile of the two ingredients previously observed (which confirmed that at skin pH they will be mainly protonated and in the cationic form) was maintained [27]. It should be remembered that a cationic character is essential for antibacterial effects because it allows electrostatic interactions with the surface of bacteria with consequent irreversible damage of membranes up to their disruption, translating in bacteria death [48,49]. Additionally, the potentiometric titrations of CP1OP2-Hgel included the titration of the NH_2_ groups of the gel, thus allowing us to determine its NH_2_ content. Potentiometric titrations of CP1OP2-Hgel were carried out as previously described [50]. The titration curves were obtained by graphing the measured pH values vs. the aliquots of HCl 0.1 N added (Figure 11a). Subsequently, from the titration data, the first derivative curve was obtaining by computing the dpH/dV values and graphing them vs. those of the corresponding volumes of HCl 0.1N (Figure 11b).

Similar to what was observed previously for copolymer CP1 [27], the titration curve of the gel composed at 6.9% wt/wt of CP1 and 1.9% wt/wt of OP2, except for some slight differences around 2 mL of HCl added, showed a very high buffer capacity up to the addition of 6–7 mL HCl 0.1 N, while the so-called jump-in pH (titration point) was visible when 8 mL of 0.1 N HCl were added (Figure 11a). Figure 11b evidences that a partial protonation started at pH values of 7–9, but complete protonation was reached at pH values of 3.52 due to the addition of 8 mL HCl 0.1 N. These results establish that the CP1OP2-Hgel at the pH of skin should be sufficiently protonated to electrostatically interact with the negatively charged surface of bacteria and boost the antibacterial effects. Table 5 collects the results concerning the content of NH_2_ groups in the gel.

### 3.8. Rheological Studies

Rheological studies are a fundamental step in the preparation of a gel formulation for skin application. Therefore, by performing rheological studies on the prepared hydrogel, the values of dynamic viscosity (η [Pa*s]) as a function of the applied shear rate (γ. [s^−1^]) were obtained. By graphing η vs. γ, we achieved the curve in Figure 12.

In Figure 12, it was observed that η decreases dramatically for a small increases in γ up to values of γ < 20, while for values of γ > 20, η was practically constant, and did not change significantly, even for great increases in γ. Collectively, the CP1OP2-Hgel is a non-Newtonian fluid. It means that the CP1OP2-Hgel does not follow the Newton’s law of viscosity given by Equation (7), where viscosity (η) is constant and independent of the shear stress (τ), and whose graphical representation is line with the constant slope (η) intercept zero [51].
τ = η × γ(7)

Non-Newtonian fluids may be Bingham plastic fluids, whose viscosity is constant, graph is linear, but intercept is >0, Bingham pseudoplastic fluids, which are modellable using the Herschel–Bulkley viscosity model and have a shear thinning behavior, or dilatant fluids, which have a shear thickening behavior [52].

To find what type of fluid the CP1OP2-Hgel was and what its behavior was when subjected to an increasing shear rate, the Cross rheological model, which is expressed by Equation (8), was fit to the data of viscosity vs. shear rate. R^2^, as well as the flow index (n) and the consistency index (α), were also determined (Figure 13, Table 6).
Log η = Log (ηo/α) − nLogγ(8)
where γ is the shear rate, μ_o_ is the viscosity when the shear rate is close to zero, n is the flow behavior index, which indicates the tendency of a fluid to shear thin or thick, and α is the consistency index, which serves as the viscosity indexes of the systems. According to the literature [52], the Cross model is obtained by graphically reporting the Log η vs. Log γ [52] (Figure 13).

In the equation of the linear tendency line of the Cross model, the intercept was Log (ηo/α), while the n value was the slope.

The equation of the linear regression obtained for the Cross mathematical model, the related correlation coefficient (R^2^), and the value of the slope and intercept are reported in Table 6, together with the computed value of α.

In particular, when n < 1 (absolute value), the fluid is shear-thinning (pseudoplastic fluid), when n = 1, the fluid is Newtonian, and when n > 1, the fluid is shear-thickening (dilating hydrogels). According to Figure 13, the linear regression of the Cross model, which fit the rheological data well, since the R^2^ value was very high (R^2^ > 0.99) provided n > 1, thus asserting the CP1OP2-Hgel is a shear-thickening, dilating hydrogel. In fact, typically, pseudoplastic fluids are relatively mobile fluids, such as weak gels and low-viscosity dispersions, while the CP1OP2-Hgel is highly viscous.

#### PCA on Rheological Data

Since it was the first time that we obtained a dilating gel, we decided to compare the rheological data of the CP1OP2-Hgel with those of some Bingham pseudoplastic gels with shear thinning behavior, previously reported in [53,54], by processing them using PCA. To this end, we collected data from rheological experiments carried out on the CP1OP2-Hgel and on the recently prepared Bingham pseudoplastic CP1-based hydrogel, made of CP1 and water, in a matrix. Then, the viscosity data of other five previously reported Bingham pseudoplastic blank hydrogels, namely PEC-D and PEC-K (the former made of pectin and Dermasoft and the latter of pectin and potassium polysorbate), CMC (made of carboxymethyl cellulose), HEC (made of hydroxy ethyl cellulose), and MC (made of methyl cellulose), were included, achieving a matrix (7 × 15) of 105 variables. The obtained matrix was then processed by PCA using PAST statistical software (paleontological statistics software package for education and data analysis, freely downloadable online at: https://past.en.lo4d.com/windows; accessed on 6 September 2022). PCA allowed us to visualize the reciprocal positions that the seven gels occupied in the scores plots of PC1 vs. PC3 (Figure 14), depending on the presence of similarities or differences in their rheological characteristics, which are included in Table 7.

Significant information was provided by PC3 on which gels appeared well separated and/or grouped in accordance mainly with their values of dynamic viscosity as a function of shear rate, of their flow behavior indices (n), and of their consistency indices (α), surely influenced by the structural and physicochemical characteristics of the gels’ components. In particular, all carbohydrate-based gels were located at negative scores on PC3, while styrene-based gels were positioned at positive scores. Additionally, the CP1OP2-Hgel was positioned at very high scores and very distant from all other samples, including CP1, thus evidencing that the CP1OP2-Hgel has different rheological behavior to those of all other gels. In fact, while CP1OP2-Hgel, characterized by values of α < 1 and n > 1 of the linear regressions obtained by fitting the rheological data to the Cross mathematical model, demonstrated a dilating, shear thickening behavior, all other gels shared a pseudoplastic shear thinning behavior characterized by values of n < 1. Additionally, all gels were not differentiated on PC1 (very similar scores), except for CP1, which was segregated at a positive score lower than other gels. We attributed this finding to the values of its consistency index (α), which was higher than those of all other gels. Within the family of carbohydrate-based gels, additional differentiations were found on PC3 on the base of the n values. In particular, PEC-D and MC with the lowest n were positioned close each other at the highest negative score and separate from PEC-K, CMC, and HEC, which formed a cluster at lower negative scores.

## 4. Conclusions

In this study, which aimed to develop a new broad-spectrum antibacterial formulation for topical administration, possibly intended for the treatment of severe skin infections caused by MDR bacteria, we reported the synthesis and physicochemical characterization of a two-component hydrogel using two previously reported bactericidal cationic polymers (CP1 and OP2) as ingredients. In this regard, to make our research more robust and relevant, we first investigated the cytotoxic behavior of CP1 and OP2 against human fibroblasts (MAIL-2). The values of the selectivity indices (SIs) obtained (up to 12) for several clinical isolates supported the progression of the study reported here. In particular, the styrene-based bactericidal copolymer CP1 was used as a bactericidal ingredient and as a thickening agent, to formulate the water-soluble bactericidal homopolymer OP2 in gel. OP2 was used equimolar with CP1. The project was based on the powerful antibacterial and bactericidal effects of the two polymers. Notably, while the antibacterial activity of CP1 and OP2 was very similar on Gram-positive MDR strains of the genus *Staphylococcus* and *Enterococcus* (with CP1 more active than OP2), it was complementary on some clinical isolates of non-fermenting Gram-negative species such as *A. baumannii* and *S. maltophylia*, with OP2 more active than CP1. Therefore, we hypothesized that a formulation holding both compounds should combine the antibacterial effects of the individual components, thus providing an antibacterial agent with a very broad spectrum of action. Our idea also derived from the physicochemical characteristic of CP1, which showed the ability to self-form gel and was successful when was used it to form a one-component Bingham pseudoplastic gel composed only of CP1 and water. This gel had a high swelling capacity, high porosity, and spreadability. No additives (such as gelling agents) or physicochemical techniques were employed to obtain the 3D gel network based on CP1 and OP2, but only water was used to disperse the two components. As a result, the CP1OP2-Hgel developed here was obtained with a low-cost procedure, accessible to the operator, and without other ingredients that could affect the biological properties of CP1 and OP2 or make the gel dangerous or incompatible with the skin for a future topical application. The chemical structure and morphology of the prepared CP1OP2-Hgel were evaluated by ATR-FTIR spectroscopy and SEM, while rheological experiments also processed by PCA established that it fits the Cross rheological model well and behaves as a shear-thickening dilating fluid. The weight loss profile over time, the equilibrium swelling ratio percentage, and the maximum swelling and porosity capacity were obtained by carrying out the necessary experiments. Weight loss experiments showed that the release of the water contained in the gel was very slow, but a quantitative release was achieved after 24 h of heating at 37°C. The release of water followed first-order kinetics, which means that it depended on the residual concentration of water in the gel. Experiments to find the equilibrium swelling ratio percentage on dry gel samples demonstrated that dried samples reform the hydrogel, swelling by 2789% in 30 min, while the porosity and maximum swelling capacity were 85–91% and 583–1033%, highlighting that the CP1OP2-Hgel, in addition to being advisable for the topical therapy of skin infections, could also promote wound healing. Finally, although the NH_2_ content in CP1 and OP2 as isolated polymers had previously been assessed, the CP1OP2-Hgel was titrated potentiometrically to determine the NH_2_ equivalents per gram of gel. Overall, due to the potent bactericidal effects of CP1 and OP2 and their favorable selectivity indices against the most common pathogens present on nosocomial objects and transmissible to humans, the CP1OP2-Hgel could become a new weapon for disinfecting the skin and for the treatment of severe skin infections.

## Figures and Tables

**Figure 1 pharmaceutics-14-02444-f001:**
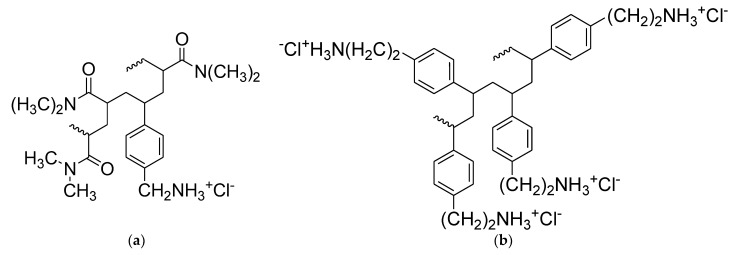
Chemical structure of CP1 (**a**) and OP2 (**b**).

**Figure 2 pharmaceutics-14-02444-f002:**
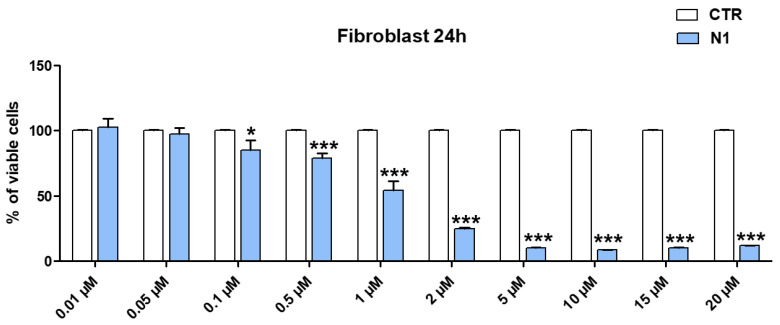
Results from dose-dependent cytotoxicity experiments carried out on MAIL-2 human cells at 24 h exposure to CP1 (N1 = CP1, CTR = control). * = *p* < 0.05, *** = *p* < 0.001.

**Figure 3 pharmaceutics-14-02444-f003:**
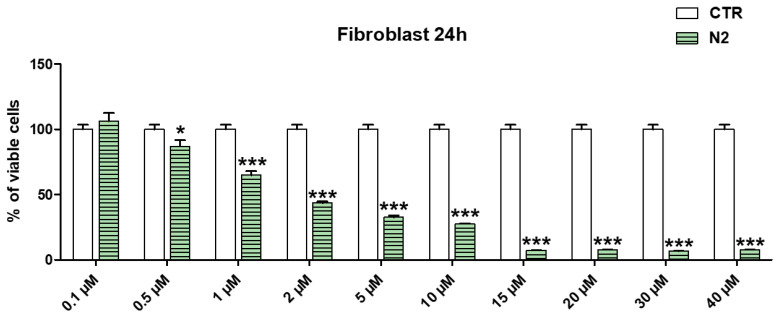
Results from dose-dependent cytotoxicity experiments carried out on MAIL-2 human cells at 24 h exposure to OP2 (N2 = OP2, CTR = control). * = *p* < 0.05, *** = *p* < 0.001.

**Figure 4 pharmaceutics-14-02444-f004:**
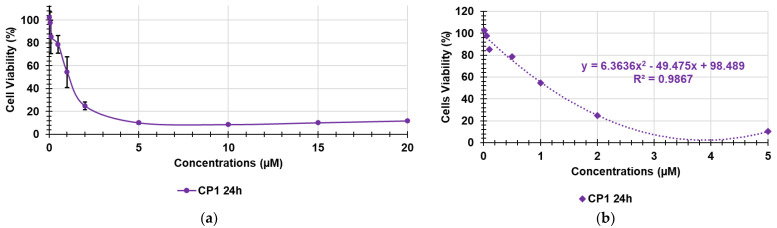
Curve of MAIL-2 cell viability (%) vs. concentrations (0.01–20 µM) of CP1 at 24 h (**a**); polynomial tendency line fitting the dispersion graph shown in Figure 4a (0.01–5 μM) (**b**).

**Figure 5 pharmaceutics-14-02444-f005:**
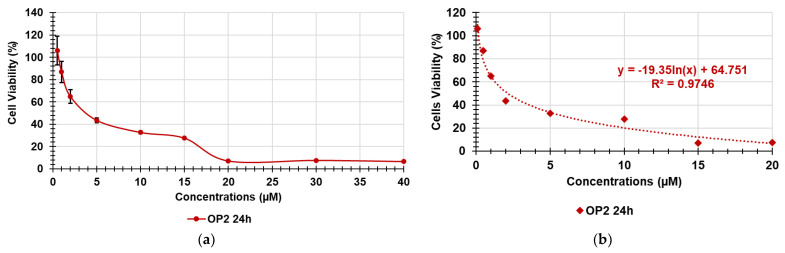
Curve of MAIL-2 cell viability (%) vs. concentrations (0.1–40 µM) of OP2 at 24 h (**a**); logarithmic tendency line fitting the dispersion graph shown in Figure 5a (0.1–20 μM) (**b**).

**Figure 6 pharmaceutics-14-02444-f006:**
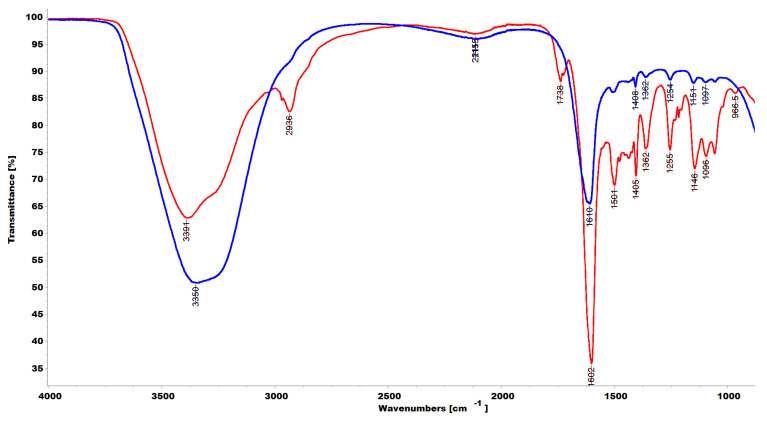
ATR-FTIR spectra of soaked CP1OP2-Hgel (blue line) and of dried CP1OP2-Hgel (red line).

**Figure 7 pharmaceutics-14-02444-f007:**
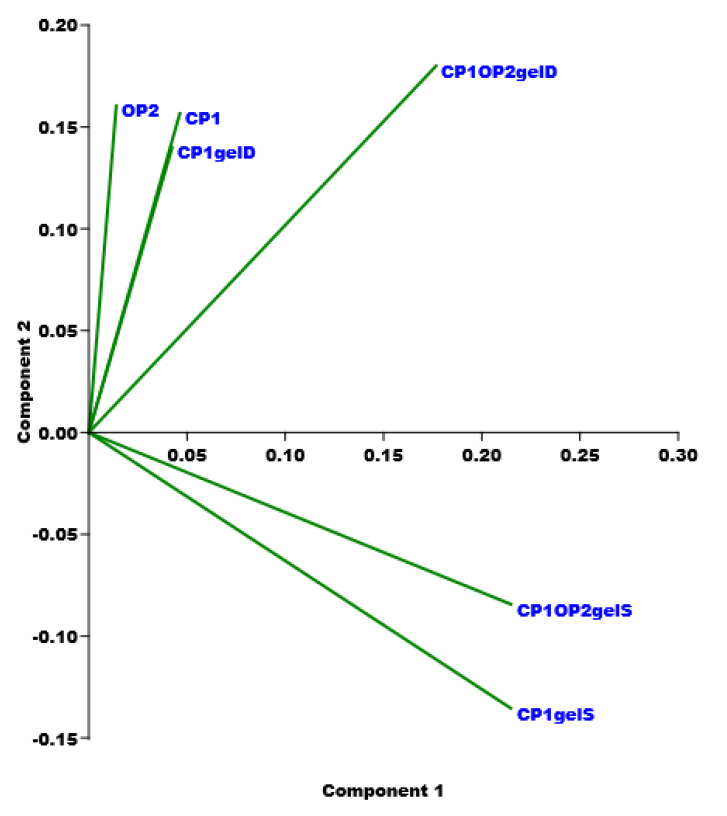
Results represented as a score plot by PCAs performed on the matrix collecting spectral data of CP1, OP2, swollen and dry CP1-based gel, as well as of swollen and dry CP1OP2-based gel developed here (PC1 vs. PC2).

**Figure 8 pharmaceutics-14-02444-f008:**
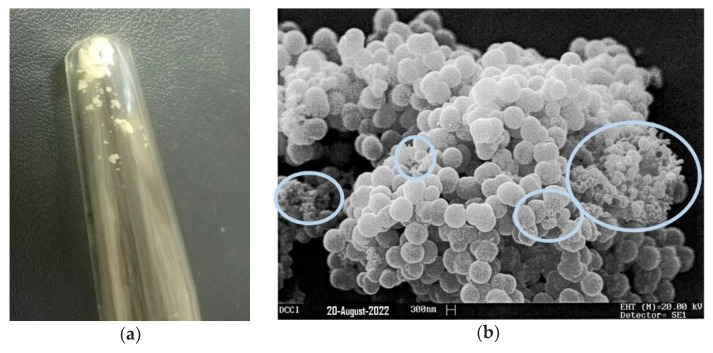
Lyophilized CP1OP2-Hgel (**a**); a representative SEM micrograph of a freeze-dried sample of CP1OP2-Hgel (**b**).

**Figure 9 pharmaceutics-14-02444-f009:**
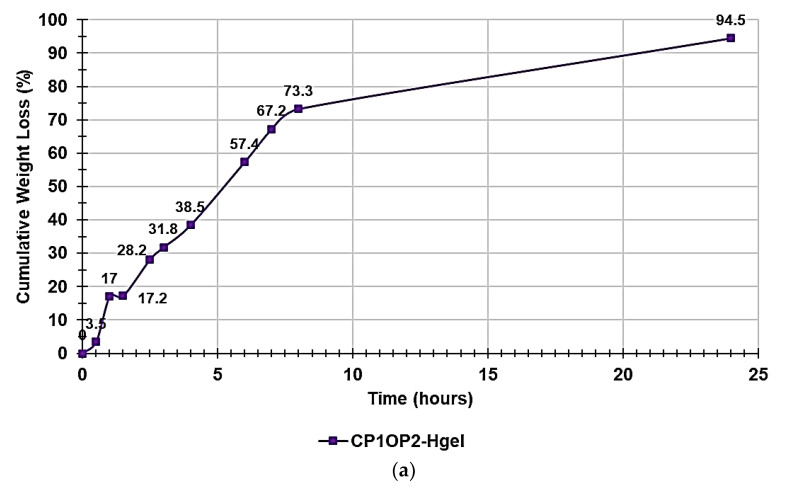
Cumulative weight loss percentage curve of the gel (**a**); first-order kinetic model fitting data of cumulative weight loss curve (**b**).

**Figure 10 pharmaceutics-14-02444-f010:**
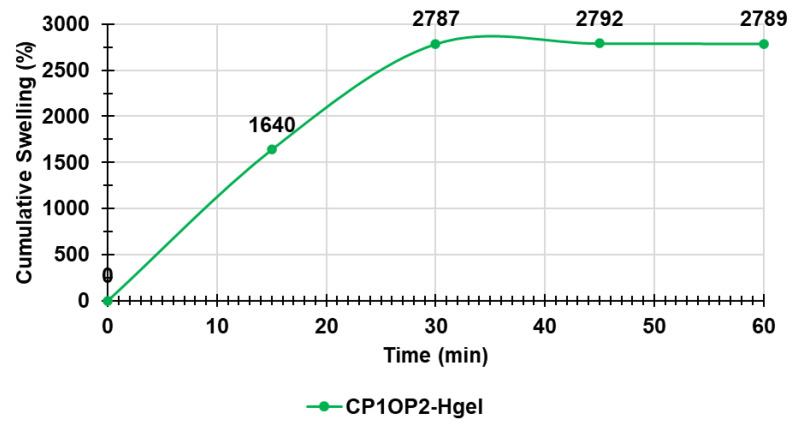
Cumulative swelling ratio percentage curve of CP1OP2-Hgel.

**Figure 11 pharmaceutics-14-02444-f011:**
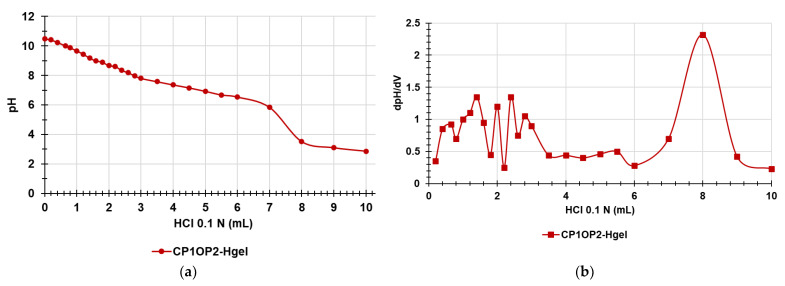
A representative potentiometric titration of CP1OP2-Hgel (**a**); first derivative (dpH/dV) of the curve in Figure 11a (**b**).

**Figure 12 pharmaceutics-14-02444-f012:**
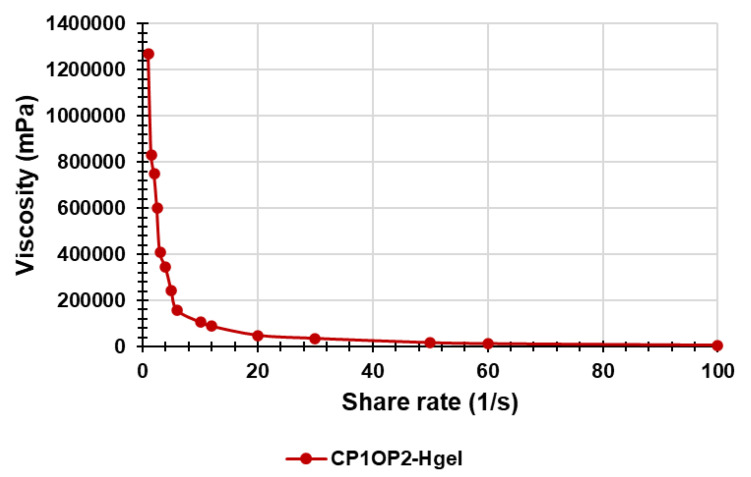
Curve of dynamic viscosity vs. shear rate.

**Figure 13 pharmaceutics-14-02444-f013:**
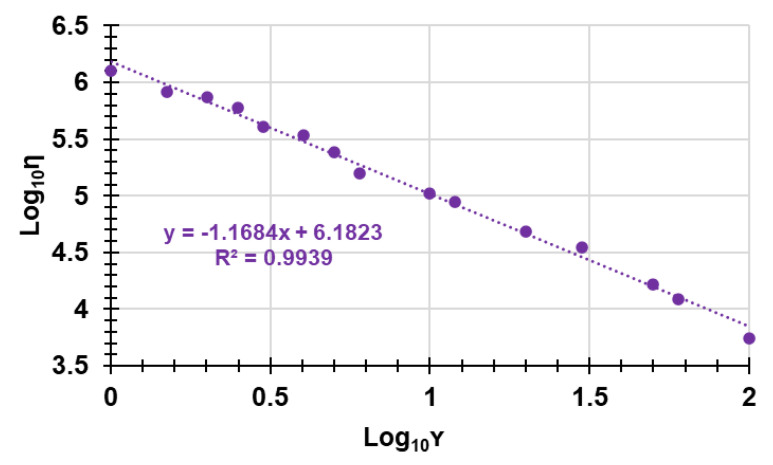
Cross rheological model.

**Figure 14 pharmaceutics-14-02444-f014:**
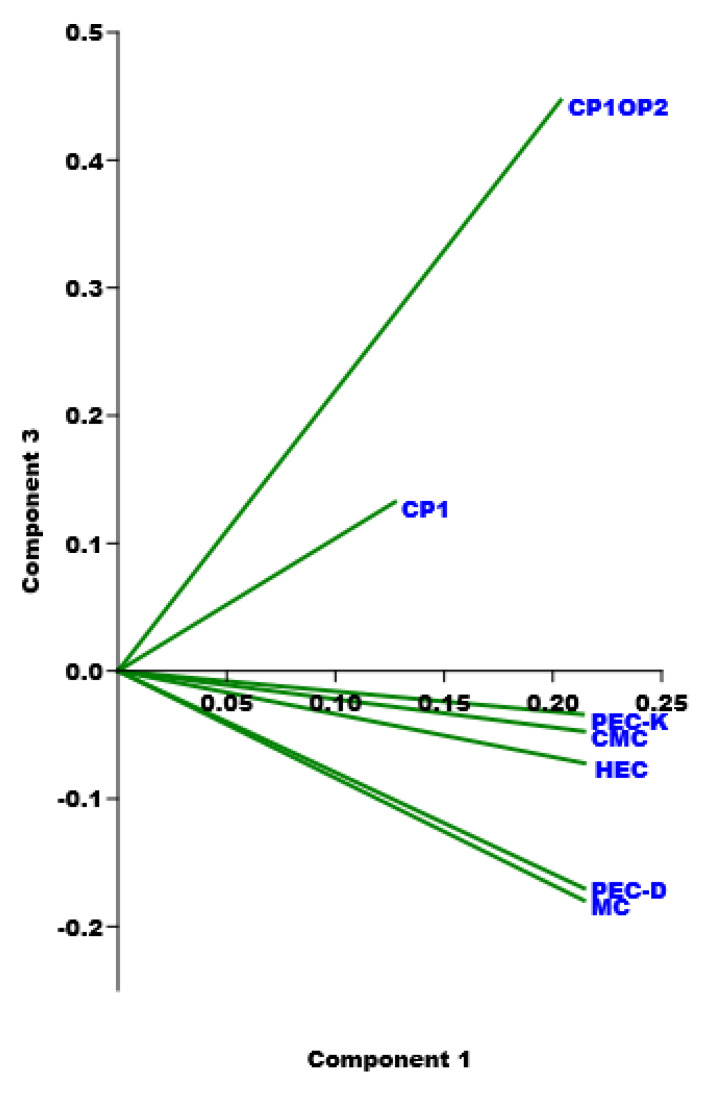
PC1 vs. PC3 by PCA experiments.

**Table 1 pharmaceutics-14-02444-t001:** Polynomial and logarithmic equations of the tendency lines, R^2^ values, LD_50_ of CP1 and OP2, as well as the relative SI ranges calculated using Equation (1).

Sample	Equations	R^2^	LD_50_ (µM)	SI	
				Gram-Positive	Gram-Negative
CP1	y = 6.3636x^2^ − 49.475x + 98.489	0.9867	1.2	1.5–12.0	1.5–3.0
OP2	y = −19.35ln(x) + 64.751	0.9746	2.1	1.5–6.0	0.75–3.0

**Table 2 pharmaceutics-14-02444-t002:** MDR clinical isolates against which CP1 and OP2 were tested and related SI values of CP1 and OP2.

Strains	CP1 (157,306) ^1^ SI	OP2 (44,514) ^1^ SI
*E. faecalis* 365 *	3.0	1.5
*E. faecalis* 450 *	1.5	1.5
*E. faecalis* 451 *	3.0	1.5
*E. faecium* 300 *	3.0	3.0
*E. faecium* 364 *	3.0	3.0
*E. faecium* 503 *,TR	3.0	3.0
*S. aureus* 18 **	3.0	3.0
*S. aureus* 187 **	1.5	3.0
*S. aureus* 195 **	3.0	3.0
*S. epidermidis* 180 ***	12.0	6.0
*S. epidermidis* 181 ***	12.0	6.0
*S. epidermidis* 363 **	6.0	3.0
*E. coli* 461	1.5	1.5
*E. coli* 462 §	1.5	1.5
*E. coli* 477 #	<1.5	1.5
*K. aerogenes* 484CAR	1.5	1.5
*K. aerogenes* 500 #	1.5	3.0
*K. aerogenes* 501CAR	3.0	1.5
*K. pneumoniae* 502 #	<1.5	1.5
*K. pneumoniae* 509 #	1.5	0.75
*K. pneumoniae* 520 #	3.0	1.5
*P. aeruginosa* 1V	1.5	3.0
*P. aeruginosa* CR	1.5	1.5
*P. aeruginosa* PY	3.0	3.0
*A. baumannii* 245	<1.5	3.0
*A. baumannii* 257	1.5	3.0
*S. maltophylia* 255	<1.5	3.0
*S. maltophylia* 280	<1.5	3.0

^1^ Mr of copolymer CP1; * denotes vancomycin-resistant isolates (VRE); TR = teicoplanin-resistant; ** denotes methicillin-resistant isolates; *** denotes resistance toward methicillin and linezolid; # denotes *K. pneumoniae* carbapenemase (KPC)-producing bacteria; § denotes New Delhi metallo-β-lactamase (NDM)-producing strains; CAR = carbapanems-resistant isolates not producing β-lactamases; *P. aeruginosa, S. maltophylia*, and *A. baumannii* are all MDR bacteria; CR = colistin-resistant isolate; PY = pyomelanin-producing strain; 1V = isolated from patient with cystic fibrosis.

**Table 3 pharmaceutics-14-02444-t003:** Experimental data for CP1OP2-Hgel preparation, details about CP1 and OP2 concentrations in the prepared hydrogel, porosity, and maximum swelling capability (by volume and by weight).

Entry	V (mL)	W (mg) (µmol)	Concentration (%wt/wt)	Vi (mL)	Wi (g)	Vf (mL)	Wi (g)	P (%)	S (%)
CP1	0.35	284.51.8	6.9 ^§^	0.35	0.2845	4.1	4.1362	N.A.	N.A.
OP2	0.25	80.51.8	1.9 ^§^	0.25	0.0805			N.A.	N.A.
CP1OP2-Hgel	0.6	365.0	8.8 ^a^	0.6	0.3650	4.1	4.1362	85.4 *	583 *
								91.2 **	1033 **

^§^ 517 µM; ^a^ 1.0 mM; * by volume; ** by weight; N.A. = not applicable.

**Table 4 pharmaceutics-14-02444-t004:** Values of coefficients of determination obtained for all the kinetic models used.

Kinetic Model	R^2^ Hydrogel
Zero-order	0.7530
**First-Order**	**0.9789**
Hixson–Crowell	0.7530
Higuchi	0.9727
Korsmeyer–Peppas	0.8938
Weibull	0.9626

**Table 5 pharmaceutics-14-02444-t005:** NH_2_ equivalents contained in the gel determined using the data obtained by its potentiometric titration.

Entry	Weight (mg)	HCl 0.1 N (mL)	NH_2_ (mmol)	mequiv._NH2_/g	µequiv._NH2_/g
CP1OP2-Hgel	284.1	8.0 ± 0.020	0.8 ± 0.0020	2.8159 ± 0.0800	2,815.9 ± 0.3
CP1+OP2	25.0	8.0 ± 0.020	0.8 ± 0.0020	32.000 ± 0.0800	32,000.0 ± 80.0

**Table 6 pharmaceutics-14-02444-t006:** Equation, correlation coefficient (R^2^), slope, and intercept values of the linear regressions obtained by fitting the rheogram data to the Cross mathematical model.

Mathematical Model	Equation	R^2^	Slope (n)	Intercept	α
Cross	y = −1.1684x + 4.0715	0.9939	1.1684	6.1823	0.83

**Table 7 pharmaceutics-14-02444-t007:** Correlation coefficient (R^2^), slope (n), intercept values, and α of the linear regressions obtained by fitting the rheogram data of gels subjected to PCA to the Cross mathematical model.

Gels	R^2^	Slope (n)	Intercept	α
CP1OP2	0.9939	1.1684	6.1823	0.83
CP1	0.9884	−0.6122	5.1341	1.95
PEC-D	0.9936	−0.4518	4.8096	1.10
PEC-K	0.9958	−0.5535	4.7539	1.11
CMC	0.9989	−0.4981	4.6053	1.04
HEC	0.9944	−0.5717	4.8498	1.12
MC	0.9972	−0.3969	4.3551	1.07

## Data Availability

All data associated to this are comprised in this manuscript.

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
