# Peer review of "Mutual Jellification of Two Bactericidal Cationic Polymers: Synthesis and Physicochemical Characterization of a New Two-Component Hydrogel"

_pharmaceutics, 2022, doi:10.3390/pharmaceutics14112444_

Round 1
Reviewer 1 Report
This manuscript reports the obtention of a bicomponent hydrogel based on cationic polymers. The obtained material showed bactericidal activity against different bacteria strains. The authors claim the obtained material may be applied for treating severe nosocomial skin infections by topical administration.
The work reported here is well written and the characterizations were well conducted. It seems this work belongs to a broader and continuous project, as the authors often cite their work submitted in Biomedicines (reference 1). The biggest problem here is that the authors did not provide the previously obtained data, forcing the reader to look for an unpublished article in another journal. I could not find these results to evaluate this work. Thus, strongly recommend providing such data related to CP1 as supporting information. Based on the exposed above, I recommend this work for publication in Pharmaceutics after minor revision.
Below, you will find some comments and suggestions.
Materials and methods
1. Line 152: The authors must include more information regarding M1 and M2 monomers. Please, provide the IUPAC names (mandatory) and give the chemical structures (optional).
2. Check the numbers, their powers, units, and equations. E.g. 4×103 cells/well instead of 4×103 cells/well. Some dots need to be removed in Equation (1). Line 213, what are the unities of numbers 4.1362 and 4.10? Please, check it throughout the manuscript.
3. The authors must give more information regarding OP2 and CP1 polymers. For example, in line 202: “OP2 (0.25 mL, 80.5 mg; 1.8 µmol)…” with such information we can calculate the molar mass (44,722.22 g/mol). However, since polymers have a molar mass (MM) distribution, the number 44,722.22 g/mol is related to the number average MM (Mn), weight average MM (Mw), or viscosity average MM (Mv)? In addition, considering the polymers that were previously synthesized by the authors, how the MM was determined? Is there some information regarding the polydispersity index (Mw/Mn)?
4. At the beginning of section 2.4, the authors state that the hydrogel was prepared by mixing 80.5 mg of OP2 and 284.5 mg of CP1 in 9.4 mL of water. Then, in line 214, the authors gave another formulation. Why are there two different formulations?
Results
5. The results shown in Figure 2b were fitted with a polynomial equation. Did the authors try to fit the data by using exponential decay or inverse functions?
6. The authors tested the bactericidal activity of CP1 and OP2 polymers. Why did not they test a mixture of CP1 and OP2? In a mixture of polymers, a synergistic or antagonistic effect may be observed being closer to the hydrogel effect.
7. Table 3. How the Wi values was obtained?
8. FTIR: Which vibration modes the bands at 1509, 1408, 1254, and 1152 cm-1 are related?
9. Figure 7 does not contribute to the discussion of the results. It can be moved to supplementary information.
10. Table 4 and Figure 8a have the same data. Please consider sending the Table for supplementary information.
11. The mathematical model of Korsmeyer-Peppas adjusts only the first 60% of the overall deswelling profile (only the linear part). Did you consider this restriction in your studies?
12. Table 6 and Figure 9 have the same data. I would suggest adding a data label to the graph and removing Table 6.
Author Response
This manuscript reports the obtention of a bicomponent hydrogel based on cationic polymers. The obtained material showed bactericidal activity against different bacteria strains. The authors claim the obtained material may be applied for treating severe nosocomial skin infections by topical administration.
We thank the Reviewer for having perfectly centred the scope of our work.
The work reported here is well written and the characterizations were well conducted. It seems this work belongs to a broader and continuous project, as the authors often cite their work submitted in Biomedicines (reference 1). The biggest problem here is that the authors did not provide the previously obtained data, forcing the reader to look for an unpublished article in another journal. I could not find these results to evaluate this work. Thus, strongly recommend providing such data related to CP1 as supporting information. Based on the exposed above, I recommend this work for publication in Pharmaceutics after minor revision.
We apologise with the Reviewer for the inconvenient, but we thought that the manuscript submitted for publication before the present one and yet under review could be available meantime the revision process of the present one could be finished. Unfortunately, for reasons unknown, the revision process of our previous study is turning out to be extremely long. Due to this situation, the Reviewer is right. As asked, we have added the still not available results related to CP1-based gel essential to better understand the present work as supporting information in a Supplementary Materials file, hoping in their fast publication. The Ref. 1 to the not yet published work has been removed.
Below, you will find some comments and suggestions.
Materials and methods
- Line 152: The authors must include more information regarding M1 and M2 monomers. Please, provide the IUPAC names (mandatory) and give the chemical structures (optional).
The IUPAC names requested have been included in main text of the revised manuscript (lines 171-172).
- Check the numbers, their powers, units, and equations. E.g. 4×103 cells/well instead of 4×103 cells/well. Some dots need to be removed in Equation (1). Line 213, what are the unities of numbers 4.1362 and 4.10? Please, check it throughout the manuscript.
The issues signalled by the Reviewer have been addressed (lines 223, 252 and 271). Additionally, all manuscript has been checked to solve similar problems.
- The authors must give more information regarding OP2 and CP1 polymers. For example, in line 202: “OP2 (0.25 mL, 80.5 mg; 1.8 µmol)…” with such information we can calculate the molar mass (44,722.22 g/mol). However, since polymers have a molar mass (MM) distribution, the number 44,722.22 g/mol is related to the number average MM (Mn), weight average MM (Mw), or viscosity average MM (Mv)? In addition, considering the polymers that were previously synthesized by the authors, how the MM was determined? Is there some information regarding the polydispersity index (Mw/Mn)?
All information requested by the Reviewer is reported in our previously published study on the antibacterial/bactericidal effects of the two polymers (Ref 25 of unrevised manuscript, Ref. 27 in the revised version). Anyway, the MM distributions used were viscosity average MMs (Mv) which were determined using the relationship
[η] = K x Ma
Anyway, to meet the Reviewer’s comments, the necessary additional information concerning OP2 and CP1 have been included in the text (lines 259-262).
- At the beginning of section 2.4, the authors state that the hydrogel was prepared by mixing 80.5 mg of OP2 and 284.5 mg of CP1 in 9.4 mL of water. Then, in line 214, the authors gave another formulation. Why are there two different formulations?
We apologise in advance with the Reviewer, but both in the word and in the PDF version of the manuscript submitted by us to Pharmaceutics, at line 214 we found Eq. (2). Anyway, we think to have understand the Reviewer’s comment and request. If the Reviewer refers to our sentence: “Accordingly, the volume of water in which CP1 and OP2 were dispersed was 3.5 mL”, we explain that 3.5 mL is not the initial volume of water added to the solids CP1 and OP2 weighted, but the volume of water maintained in the gel formulation (i.e. that one absorbed by CP1 and OP2) once the not absorbed water was removed (i.e. 4.1 mL – 0.6 mL = 3.5 mL). Anyway, we agree with the Reviewer that the original sentence is unclear. The sentence has been changed (lines 272-274).
Results
- The results shown in Figure 2b were fitted with a polynomial equation. Did the authors try to fit the data by using exponential decay or inverse functions?
We have fitted the functions suggested by the Reviewer and even others, but the R2 values resulted in any case less high.
- The authors tested the bactericidal activity of CP1 and OP2 polymers. Why did not they test a mixture of CP1 and OP2? In a mixture of polymers, a synergistic or antagonistic effect may be observed being closer to the hydrogel effect.
The Reviewer is right concerning the possible changes in the antibacterial effects of CP1 and OP2 once mixed together. However, as better evidenced in the modified title of the present manuscript, the scope of this study, was to formulate CP1 and OP2 as gel topically administrable and to report its full physicochemical characterization. The antibacterial effects and non-cytotoxicity to human cells of the gel formulation reported here are currently undergoing and will be reported in a following work, which will also report the assessment of the antibiofilm properties of CP1OP2-Hgel. We think that such experiments directly on the gel could be more relevant that experiments made using the simple physical mixture of CP1 and OP2, that only would mimic the real gel. We ask kindly the Reviewer, to be patient a little longer to know these results.
- Table 3. How the Wi values was obtained?
The requested information has been better evidenced, by inserting in Section 2.4 Vi and Wi, as well as Vf and Wf near the correspondent numbers (lines 259 and 271).
- FTIR: Which vibration modes the bands at 1509, 1408, 1254, and 1152 cm-1 are related?
The requested information has been added (lines 524-525).
- Figure 7 does not contribute to the discussion of the results. It can be moved to supplementary information.
As requested, Figure 7 has been moved to SM where it appears as Figure S5.
- Table 4 and Figure 8a have the same data. Please consider sending the Table for supplementary information.
The Reviewer is right. Table 4 has been moved to SM, where it compares as Table S1, while Figure 8 (now Figure 7) was maintained in the main text of the revised manuscript.
- The mathematical model of Korsmeyer-Peppas adjusts only the first 60% of the overall deswelling profile (only the linear part). Did you consider this restriction in your studies?
We apologise in advance to the Reviewer, but in the present work we have reported that the first order mathematical model fits the overall deswelling profile of CP1OP2-gel (R2 = 0.9789), while Korsmeyer-peppas kinetic model fitted that of CP1-gel previously prepared by us, and whose data as previously suggested by the Reviewer for other ones, have been included in SM in Figure S6 and Table S2.
- Table 6 and Figure 9 have the same data. I would suggest adding a data label to the graph and removing Table 6.
As asked, data labels have been included in Figure 8 (revised manuscript) and Table 6 has been removed.
Reviewer 2 Report
The article reports a bicomponent Hydrogel formulation by using two bacteri- 2 cidal cationic polymers, which physiological properties have been studied in detail, while some major revisions need before acceptance.
1. the chemical structure of CP1 and homopolymer OP2 should be drawn in text.
2. Fig 2. why the cell viability stayed constant in a wide concentration, which should be discussed in text. and n value for each point should be given.
3. Fig 3 should be removed from text.
4. Fig 4 should be improved to remove the inset lines.
5. Figure 11, the mechanical modulus dependent on shear rate should be given together.
6. all figures should be carefully improved to be better in text.
Author Response
The article reports a bicomponent Hydrogel formulation by using two bactericidal cationic polymers, which physiological properties have been studied in detail, while some major revisions need before acceptance.
- the chemical structure of CP1 and homopolymer OP2 should be drawn in text.
We thank a lot the Reviewer for his suggestion. The chemical structure of CP1 and OP2 have been included in the Introduction Section, as Figure 1a and 1b. The order of the subsequent Figures has been updated
- Fig 2. why the cell viability stayed constant in a wide concentration, which should be discussed in text. and n value for each point should be given.
The requested information has been included in the main text after the original Figure 2 (Figure 3 in the revised version). Please, see lines 431-437).
- Fig 3 should be removed from text.
As requested, Figure 3 has been removed from the main text and inserted in the new Supplementary Materials file, where it appears as Figure S4.
- Fig 4 should be improved to remove the inset lines.
As requested, Figure 4 has been improved by removing the grid.
- Figure 11, the mechanical modulus dependent on shear rate should be given together.
We apologise in advance with the Reviewer, but the mechanical modulus (we think the Young’s modulus) does not depend on the shear rate but on the shear stress, and in particular is given by the ratio between shear stress and shear strain, being the latter provided by a tensiometer and not by a viscosimeter, as that used by us. Particularly, Young's modulus (i.e. the modulus of elasticity in tension or compression) is a mechanical property that measures the tensile or compressive stiffness of a solid material when the force is applied lengthwise. It quantifies the relationship between tensile/compressive stress and axial strain (proportional deformation) in the linear elastic region of a material. Anyway, to avoid confusion, “mechanical properties” have been replaced with “rheological properties” or similar expressions along all manuscript.
- all figures should be carefully improved to be better in text.
As requested, all the Figures have been improved.
Reviewer 3 Report
Dear Authors,
Your paper is very interesting, and the topic is well argued, presenting a worldwide problem that needs solving by researchers and scientists also.
You did excellent work!
Author Response
Dear Authors,
Your paper is very interesting, and the topic is well argued, presenting a worldwide problem that needs solving by researchers and scientists also.
You did excellent work!
We thank the Reviewer for his appreciations, and for having defined our work as excellent.
Reviewer 4 Report
The coauthors describe the development of a new hydrogel using tow bactericidal cationic polymers.
Overall, English Grammar, synatax, and spell check revisions required.
Sect 2.2.1 Please add tissue culture methods and not reference an unpublished manuscript.
Sect. 2.2.2 Please add viability methods and not reference an unpublished manuscript. I have no idea what type of viability assay you performed.
Selectivity index is used to compare antiviral activity and human cell biocompatibility. Why have you not performed an MIC paired with a cytotoxicity assay, like MTT or XTT?
Fig.1 4h exposure is short. An MTT is usually over 48h.
The antibacterial efficacy and noncytotoxicity to human cells is not established.
Author Response
The coauthors describe the development of a new hydrogel using tow bactericidal cationic polymers.
Overall, English Grammar, synatax, and spell check revisions required.
According to the suggestions of the Reviewers, we have asked Professor Deirdre Kantz, English teacher mother tongue working for University of Genoa and Pavia (Italy), to revise further our manuscript to improve the English grammar and syntax. As the Reviewer can observe, the work has been extensively revised also in this sense.
Sect 2.2.1 Please add tissue culture methods and not reference an unpublished manuscript.
We apologise with the Reviewer for the inconvenient, but we thought that the manuscript submitted for publication before the present one and yet under review could be available meantime the revision process of the present one could be finished. Unfortunately, for reasons unknown, the revision process of our previous study is turning out to be extremely long. Due to this situation, the Reviewer is right, and we have added the fibroblast culture methods in the experimental part (lines 200-221) and created a Supplementary Materials file including Figures and information reported in the still unpublished manuscript.
Sect. 2.2.2 Please add viability methods and not reference an unpublished manuscript. I have no idea what type of viability assay you performed.
We apologise with the Reviewer for the inconvenient. In this case, we have added the viability methods including information concerning the viability essay performed in lines 222-232.
Selectivity index is used to compare antiviral activity and human cell biocompatibility. Why have you not performed an MIC paired with a cytotoxicity assay, like MTT or XTT?
We make kindly note to the Reviewer that our work deals with the antibacterial effects of our polymers and not with antiviral activities. Additionally, we have computed SI values using MIC values and values of LD50 obtained by a fluorescence-based proliferation and cytotoxicity assay (CyQUANT® Direct Cell Proliferation Assay (Thermo Fisher Scientific, Life Technologies, MB, Italy) according to the manufacturer’s instructions. This information is now reported in main text (lines 227-232).
Fig.1 4h exposure is short. An MTT is usually over 48h.
We thank the Reviewer for his comment which has given us the possibility to improve our work with additional details concerning our choice to assess the cytotoxicity of CP1 and OP2 at a time of exposure of 4 hours, which is shorter than the usual times of an MTT text. According to data reported in literature, concerning the evaluation of in vitro toxicity of compounds thought for topical administration, cell viability is often determined after short times of even 25 min of exposure (Kahook, M. Y., & Ammar, D. A. (2010). In Vitro Toxicity of Topical Ocular Prostaglandin Analogs and Preservatives on Corneal Epithelial Cells. Journal of Ocular Pharmacology and Therapeutics, 26(3), 259–263. doi:10.1089/jop.2010.0003). Presently, we decided to evaluate it at 4 hours of exposure, because CP1 and OP2 displayed bactericidal activity at 4 and 2 hours of exposure, respectively (Ref. 27, revised manuscript). Such specifications and a new reference have been added in the text (lines 406-413).
The antibacterial efficacy and noncytotoxicity to human cells is not established.
We apologise in advance to the Reviewer, but his comment isn’t totally clear to us. If the Reviewer means that the antibacterial effects and non-cytotoxicity to human cells of the gel formulation here reported and physico-chemically characterized have been not assessed, we underline that the evaluation of such activities was not in the scope of the present work. Such evaluations are currently undergoing, and the results will be reported in a following work which also will report the assessment of the antibiofilm properties of CP1OP2-Hgel. Anyway, to evidence better the aims and topics of this study, the title has been modified.
Round 2
Reviewer 2 Report
It has been well revised.
Author Response
It has been well revised.
We thank a lot the Reviewer for his positive comment.
Reviewer 4 Report
The reference to an Ocular journal is not applicable for in vitro skin cytotoxicity. The material developed appears to kill both bacterial and mammalian cells and therefore in my view has no potential as an antibacterial hydrogel. Conventional testing as recommended remains to be performed.
Author Response
The reference to an Ocular journal is not applicable for in vitro skin cytotoxicity.
According to the suggestion of the Reviewer, reference 31, regarding an Ocular Journal and judged by the Reviewer not applicable in our case has been removed.
The material developed appears to kill both bacterial and mammalian cells and therefore in my view has no potential as an antibacterial hydrogel. Conventional testing as recommended remains to be performed.
As asked by the Reviewer conventional testing have been performed. We repeated the cytotoxicity experiments at 12 and 24 hours. From results, significant differences were not observed, but, unexpectedly, values of LD50 slightly higher for both compounds (hence supporting the potential of our hydrogel as an antibacterial formulation for topical application) were observed at 24 hours compared to 4 hours, highlighting that the cytotoxicity of CP1 and OP2 is only dose-dependent and not dependent on exposure time. Established this, the data obtained at 24 hours concerning both the cells viability percentage and the values of LD50 were reported, and the latter were used to recalculate the SI values. We hope that the Reviewer could now be satisfied with our work. To this end, new Figures have been included in the main text and data obtained for OP2 in the previous experiments at 4 hours have been moved in the SM close to those obtained with CP1. Please, see Sections 2.2.2. and 3.1.